# The Role of the Insular Cortex in Pain

**DOI:** 10.3390/ijms24065736

**Published:** 2023-03-17

**Authors:** Charalampos Labrakakis

**Affiliations:** 1Department of Biological Applications and Technology, University of Ioannina, 45110 Ioannina, Greece; clabrak@uoi.gr; 2Institute of Biosciences, University Research Center of Ioannina (URCI), 45110 Ioannina, Greece

**Keywords:** insula, neuropathic pain, plasticity, comorbidities, descending modulation

## Abstract

The transition from normal to chronic pain is believed to involve alterations in several brain areas that participate in the perception of pain. These plastic changes are then responsible for aberrant pain perception and comorbidities. The insular cortex is consistently found activated in pain studies of normal and chronic pain patients. Functional changes in the insula contribute to chronic pain; however, the complex mechanisms by which the insula is involved in pain perception under normal and pathological conditions are still not clear. In this review, an overview of the insular function is provided and findings on its role in pain from human studies are summarized. Recent progress on the role of the insula in pain from preclinical experimental models is reviewed, and the connectivity of the insula with other brain regions is examined to shed new light on the neuronal mechanisms of the insular cortex’s contribution to normal and pathological pain sensation. This review underlines the need for further studies on the mechanisms underlying the involvement of the insula in the chronicity of pain and the expression of comorbid disorders.

## 1. Introduction

Pain serves an important adaptive biological function. It is fundamental for limiting injury from potentially damaging stimuli by initiating reflexive and defensive behaviors and promoting the healing of tissue damage by mobilizing self-caring responses. It also is an integral part of learning and memory functions. The sensation of pain is a complex phenomenon that, besides a sensory component, also encompasses affective/motivational aspects as well as cognitive aspects. This complexity corresponds to the diverse cortical and subcortical structures that are thought to mediate different aspects of pain [1]. However, no single brain area is devoted to processing pain information exclusively; instead, the combined activity and interaction between these areas is believed to generate the experience of pain [2]. Primary and secondary sensory cortices are activated during pain and are implicated in processing the location, intensity, and quality of pain, i.e., the sensory–discriminative aspect [3]. On the contrary, affective–motivational aspects of pain, like its unpleasantness and the negative mood it evokes, are assessed in the parabrachial nucleus, the cingulate cortex, the amygdala, and the anterior insula. The prefrontal cortex and other limbic structures are also involved in the information transfer related to the cognitive component of pain [4,5] which encompasses attention, anticipation of pain, and memory of past experiences. These areas comprise a brain network that might also contribute to the transition from acute pain to chronic pain [6]. Indeed, an extensive reorganization in brain activity in chronic pain has been described [7]. This involves alterations in activity, cortical thickness, and gray matter density in several brain regions, including somatosensory, motor, insular, and prefrontal cortices, as well as in the thalamus, the amygdala, the basal ganglia, and the hippocampus [8]. Similar changes in synaptic and firing properties in different brain regions have also been described in rodent models of chronic pain [9,10,11,12,13,14].

A key cortical area that plays an important role in both physiological and possibly pathological pain is the insular cortex [15,16]. From all the brain areas involved in pain processing, only the insula and the secondary somatosensory cortex generate pain when stimulated [17]. In addition, insular lesions lead to nociceptive deficits [18]. Because the insula is integrating sensory with emotional and cognitive processes and is involved in aversive motivational salience [19], it is plausible that persistent alterations in its circuitry and synaptic physiology might underlie chronic pain and the emergence of comorbidities. However, research on the role of the insular cortex in pain and its pathology is still limited. Here we review the function of the insula and its role in pain with a focus on preclinical pain models.

## 2. Insular Cortex Function

The insula is a large cortical area where multimodal inputs from different brain areas converge. In primates, it is located within a fold of the lateral sulcus, while in rodents, it lies at the lateral surface of the brain. It can be divided into a posterior and an anterior part. These display differential connectivity to other brain areas [20]. It also consists of cytoarchitecturally diverse areas, namely a six-layered granular area, a layer IV-lacking agranular area, and an intermediate dysgranular area. The insula is believed to be a hub for the integration of multimodal information [19]. Thus, several functions and processes have been ascribed to the insula. For instance, parts of the insula function as the primary gustatory cortex [21]. It is also proposed that the insula functions as the thermosensory cortex [22] and, additionally, as the visceral sensory cortex [23]. These and other functions encompass information of the internal physiological and homeostatic condition of the body, which demonstrates that the insula serves as the primary interoceptive cortex [24,25]. As a representation of the body’s state, the insula is also believed to form the neurobiological substrate for the recognition and awareness of one’s “self”; indeed, seeing images of one’s self activates the insular cortex [26]. This is also supported by insular activation during body control awareness, heartbeat awareness, and emotional awareness [25]. The strong interconnection between the insular cortex and the limbic system indicates an essential role of the insula in emotion. Recent studies demonstrate its involvement in aversive behaviors [27], including fear [28] and anxiety [29,30]. This role also includes positive emotions, such as happiness [19,25]. Several cognitive functions have also been ascribed to the insular cortex. Preclinical studies in rats show its involvement in learning [31,32], especially aversive [33] and affective learning [34]. This involvement in memory is dependent on the saliency of the stimulus, consistent with the role of the insula as part of the salience network [35]. Similarly, insular neurons respond to relevant physiological cues in a predictive capacity of a future state [36], hence implicating the insula in anticipation of future states, prediction, prediction error computation [37], and risk estimation [19]. Conclusively, the insular cortex is acting at the interface between physiological and emotional states, with the outcome of insular function influencing motivated behaviors. Pathological conditions related to these functions have also been associated with the insular cortex [37,38,39]. Altered insular function has been described in anxiety disorders [30]; likewise, studies have shown that insular cortex gray matter volume and connectivity are altered in major depression [40,41]. Hypoactivity of the insular cortex is consistently encountered in autism disorders [42], and insula dysfunction might also contribute to schizophrenia [43]. Furthermore, the insula has been implicated in obesity and addiction [44,45].

## 3. Association of the Insular Cortex Function and Pain in Human Studies

Accumulating evidence supports a crucial role of the insular cortex in pain processing. Several human imaging and electrophysiological studies (Table 1) show activation of the insular cortex after a noxious stimulus [46,47,48]. In fact, it is the brain area found activated most frequently in fMRI pain studies [3]. Multi-site intracortical recordings show that the insula cortex is one of the brain regions that activates with the earliest latencies to nociceptive stimulation, evidence that insular processing of pain occurs in parallel to other brain regions [49]. Notably, insular activation correlates well with the intensity of the noxious stimulus [50,51,52], indicating involvement in intensity coding. Its activation also correlates with the perceived magnitude of pain [53], an important finding as it links subjective pain perception and insular cortex function directly. The nociceptive specificity of insular activation is also supported by the findings that thermal nociceptive but not innocuous stimuli cause gamma band oscillation enhancement in the insula [54,55]. Accordingly, electrical stimulation of the insula in humans elicits a sensation of pain [17,56]. This has not been described for any other cortical area, except the secondary somatosensory cortex. Consistently, lesions of the insula alter pain perception [57]. Higher pain ratings have been described in lesioned individuals [58], as well as a lack of motor withdrawal to painful stimuli, absent emotional response, and indifference to pain, termed pain asymbolia [59]. In strokes that involve the posterior insula, dissociated sensory loss with neuropathic pain has been observed [18]. Similarly, insular lesion-mediated seizures are accompanied by intensely painful attacks [60]. Clinical studies also emphasize the role of the insular cortex in chronic pain [61], showing a relationship between alterations in insular activity [62], structure [63], and pain chronification. In chronic lower back pain patients, for instance, changes of the insular connectivity with the somatosensory [64] and the medial prefrontal [65] cortices are found. On the other hand, pharmacological treatment of chronic pain reverses the altered insular cortex connectivity in chronic pain patients [66]. Increased excitability of the insula as the cause of pain hypersensitivity in pain syndromes is also supported by the finding that fibromyalgia patients have higher glutamate levels in the posterior insula, and these levels associate well with pain thresholds [67].

Pain activation of the insula follows somatotopic organization [68,69], but whether this relates to location coding is still unknown. Different roles in the processing of pain have been suggested for the posterior vs. the anterior part of the insula. Stimulation and imaging studies show that the posterior insular cortex (pIC) is involved mainly in coding pain intensity [70,71]. This is further confirmed in studies of insular lesions, while individuals with lesions in the pIC show higher pain intensity ratings, this is not the case with individuals with lesions in the anterior insular cortex [72]. The anterior insular cortex (aIC) function is mostly correlated with the emotional processing of pain [73,74], cognitive evaluation of pain [75], and empathy for pain [76,77]. In a recent study in human subjects, it was shown that the insular cortex codes the perceived intensity of empathic pain [78]. The pIC and aIC are also temporarily segregated; pain signals arrive in the insula first, in the posterior part, while processing in the anterior insula follows with some delay, indicating a posterior to anterior flow of information [46]. Pathological pain also seems to distinctively engage the different insula subregions, since insula activation shifts from the sensory (pIC) insula to the affective (aIC) insula in chronic pain [79].

Recent clinical studies have targeted the insular cortex as a potential site for deep brain stimulation-based therapeutic intervention in chronic pain. Transcranial magnetic stimulation (TMS) of the posterior operculo-insular cortex in healthy subjects increased Aδ-fiber-depending heat threshold, but not innocuous vibrotactile perception [80]. In central neuropathic pain patients, repeated TMS of the posterior insula increased thermal threshold without, however, affecting neuropathic pain scores [81]. On the contrary, in a study from the same group involving peripheral neuropathic pain patients, repeated TMS had a significant analgesic effect, which, however, was short-lasting [82]. In a pilot study in epileptic patients, electrical stimulation of the anterior insula increased heat pain thresholds [83]. It is therefore apparent that the insular cortex could become a significant site for nonpharmacological therapies for chronic pain management.

**Table 1 ijms-24-05736-t001:** Overview of human studies on the role of the insula in pain.

Subject Condition	Brain Region	Experimental Procedure	Measure	Nociceptive Stimulus	Outcome	Ref.
Healthy	Anterior and/or posterior insula	PET scan	Cerebral blood flow	Thermal	Activation	[47,50]
Healthy	Anterior and/or posterior insula	fMRI	BOLD	Thermal	Activation	[51,53,69,75]
Healthy	Anterior or posterior insula	fMRI	BOLD	Electrical pain	Activation	[52,74]
Healthy	Anterior and posterior insula	fMRI	BOLD	Muscle and cutaneous hypertonic solution injection	Activation	[68]
Healthy	Posterior insula	fMRI	BOLD	1% capsaicin cream	Activation	[71]
Healthy	Anterior and posterior insula	fMRI	BOLD	Electrical pain and empathic pain	Activation	[76]
Healthy	Anterior insula	EEG	EEG potentials	Thermal (laser)	Activation	[73]
Healthy	Posterior insula	TMS	Patient rating	Thermal	Pain threshold increase	[80]
Epileptic	Posterior and anterior insula	Electrical recording	Evoked cortical potentials	Thermal (laser)	Insula potentials	[46,49]
Epileptic	Posterior and anterior insula	Electrical recording	Evoked cortical potentials	Thermal (laser)	Insula potentials and enhanced gamma band oscillations	[54,55]
Epileptic	Posterior insula	Electrical stimulation	Behavioral pain responses or patient rating	Intracortical stimulation	Pain sensation	[17,56,70]
Epileptic	Anterior and posterior insula	Electrical recording	Intracortical activity	Empathic pain	Activity increase	[78]
Epileptic	Anterior insula	Electrical stimulation	Patient rating	Thermal	Pain thresholds increase	[83]
Insula lesion	Posterior insula	No procedure	Patient rating	Thermal	Spontaneous pain and dissociated sensory loss	[18,57]
Insula lesion	Posterior insula	No procedure	Patient rating	Thermal	Higher pain intensity ratings	[58]
Insula lesion	Anterior and/or posterior insula	No procedure	Behavioral motor response	Thermal, pinprick, heavy pressure	Pain asymbolia	[59]
Insula lesion	Anterior and posterior	No procedure	Patient rating	Thermal, mechanical	Lesion location dependent pain sensitivity alterations	[72]
Insula lesion	Anterior insula	No procedure	Patient rating	Empathic pain	Altered empathic pain perception	[77]
Insular lesion and epilepsy	Posterior insula	Electrical recording and stimulation	Behavioral pain responses	Spontaneous and intracortical stimulation	Pain sensation during ictal discharges and after stimulation	[60]
Neuropathic pain/painful mononeuropathy	Anterior insula	PET	Cerebral blood flow	No stimulation	activation	[62]
Neuropathic pain (trigeminal)	Anterior and posterior insula	MRI	Voxel-based morphometry	No stimulation	Gray matter volume changes	[63]
Chronic low back pain	Anterior insula	fMRI	BOLD	Physical maneuver-induced pain exacerbation	Increased insular functional connectivity to S1	[64]
Chronic back pain	ND	fMRI	BOLD	No stimulation	Increased insular functional connectivity to mPFC	[65]
Chronic low back pain	Anterior and mid insula	MRI	DTI	No stimulation	Decreased insular connectivity to dlPFC, increased connectivity after pharmacological treatment	[66]
Back pain (chronic or subacute)	Anterior Insula	fMRI	BOLD	Spontaneous pain	Activation	[79]
Fibromyalgia	Posterior insula	MRI	H-MRS	No stimulation	Increased glutamate levels	[67]
Chronic central neuropathic pain	Posterior insula	TMS	Patient rating	Thermal	Increased threshold	[81]
Peripheral neuropathic pain	Posterior insula	TMS	Patient rating	No stimulus	Pain score improvements	[82]

BOLD: Blood oxygenation level-dependent, dl/mPFC: dorsolateral/medial prefrontal cortex, DTI: diffusion tension imaging (f) MRI: (functional) magnetic resonance imaging, H-MRS: proton magnetic resonance spectroscopy, PET: positron emission tomography, S1: primary somatosensory cortex, TMS: transcranial magnetic stimulation.

## 4. Preclinical Studies on the Role of the Insula in Pain

Human studies have provided a wealth of information on the association of the insula with pain sensation; however, the cellular and synaptic mechanisms involved in this association are still unclear. Preclinical animal models with the aid of state-of-the-art techniques are now starting to fill the gap. Animal research on the role of the posterior insula in pain has been limited (Table 2). In vivo imaging of individual cells of the mouse pIC show that 28% of the cells respond to electrical tail shocks/pain stimuli [27]. In the same study, it was shown that optogenetic stimulation of pIC neurons was aversive, indicating that activation of pain responsive neurons of the pIC might lead to affective behaviors. Optogenetic activation of afferents from the mid-cingulate cortex in the mouse pIC induces mechanical pain hypersensitivity [84], and similarly, activation of anterior cingulate afferents in the pIC leads to mechanical and also thermal hypersensitivity [85]. In agreement with these studies, it was shown that lesions of the granular posterior insular cortex in neuropathic rats attenuated mechanical allodynia [86], pharmacological inactivation of the pIC in neuropathic macaques reduced thermal hypersensitivity [87], and optogenetic silencing of pIC pyramidal neurons in mice inhibited capsaicin-induced hypersensitivity [84]. Thus, activation of pIC during tonic and chronic pain signals increased pain sensitivity. Indeed, neurotransmission in the pIC in neuropathic pain models is altered [88], as shown by the increased expression of muscarinic M2 receptors and other cholinergic transmission components. These alterations, though, seem to enhance endogenous analgesia, since injections of M2 agonist in the pIC had analgesic effects [88]. The role of the pIC during baseline pain transmission is less clear; some evidence suggests that the pIC might not be involved in intensity coding during acute pain. Optogenetic inhibition of pIC neurons does not change hot plate thresholds [27], neither do lesions of the pIC affect basal mechanical sensitivity [86]. Other studies, however, report increased mechanical and thermal thresholds after optogenetic inactivation of ACC afferents in the pIC [85]. This could imply that the role of the pIC in acute pain is input and context dependent.

In the aIC, increase of the extracellular GABA concentration with vigabatrin or GAD overexpression in rats decreases heat paw-withdrawal sensitivity [89], indicating a relation between basal aIC activity and acute heat thresholds. In other studies, however, lesions of the aIC had no effect in acute heat pain responses [90]. Nevertheless, the lesions attenuate tonic inflammatory and neuropathic pain. Similarly, aIC activity is increased in chronic pancreatitis pain [91]. Enhancement of aIC excitability in the mouse nerve ligation chronic pain model is the outcome of the GluA1 AMPA receptor subunit phosphorylation and upregulation [92]. Indeed, inhibition of glutamatergic transmission with injections of antagonists in the aIC is analgesic [91,92]. Thus, plastic changes in the aIC might be involved in chronic pain hypersensitivity. This agrees with increased NMDA receptor expression in aIC synapses of neuropathic mice [93]. Plastic changes in posterior and anterior insula have also been described in neuropathic rats using fMRI imaging [94]. Consequently, the aIC has been implicated as a site of antinociceptive action of several modulatory transmitters. Microinjection of morphine into the aIC has an antinociceptive effect in the formalin test [95]. Furthermore, opioid receptor antagonist injection in the aIC reduced the antinociception of systemic application of morphine, which suggests that the aIC is a major contributor to opioid antinociception. Dopaminergic transmission in the aIC is also involved in pain modulation, as injection of reuptake inhibitors in the anterior insula is antinociceptive [96]. Moreover, local injection of a D1-receptor antagonist or a D2-receptor agonist in a rat neuropathic pain model produces antinociception [97]. However, in the formalin test, the same D1-receptor antagonist shows pronociceptive action [96]. Thus, the role of dopamine in the aIC might differ between chronic neuropathic pain and acute tonic pain. Oxytocin injections in the aIC also have antinociceptive effects [98]. Finally, similar to the human insula, the aIC plays a role in empathic pain in mice [99,100]. Uninjured mice acquire mechanical hypersensitivity after observing neuropathic littermates and this empathic hypersensitivity depends on aIC connections to the amygdala.

**Table 2 ijms-24-05736-t002:** Overview of pre-clinical studies on the role of the insula in pain.

Subregion	Species	Pain Model	Insula Manipulation	Pain Measure	Noxious Stimulus	Outcome	Ref.
Posterior Insula	Mouse	Naïve	In vivo calcium imaging	Cell activity	Electric tail shock	Activation	[27]
Posterior Insula	Mouse	Naïve	Optogenetic Excitation of MCC inputs activation	Paw withdrawal	Mechanical	Increased sensitivity	[84]
		Capsaicin	Optogenetic inhibition of pyramidal neurons			Decreased sensitivity	
Posterior Insula	Mouse	Sham & 6-OHDA	Optogenetic excitation of ACC inputs	Paw withdrawal	Mechanical & Thermal	Increased sensitivity	[85]
			Optogenetic inhibition of ACC inputs			Decreased sensitivity	
Posterior Insula	Rat	Neuropathic	Lesion	Paw withdrawal	Mechanical	Reduced sensitivity	[85]
Posterior Insula	Macaque	Naive	Pharmacologic inactivation (muscimol)	Hand withdrawal	Thermal	Decreased sensitivity	[87]
Anterior Insula	Rat	Naive	Pharmacologic & genetic inactivation	Paw withdrawal	Thermal	Decreased sensitivity	[89]
Anterior Insula	Rat	Naive	Pharmacologic inactivation (morphine)	Formalin score	Chemical (Formalin)	Reduced score	[95]
				Spinal cFos		Decrease in expression	
				Spinal electro-physiology		Decrease of firing	
Anterior Insula	Rat	Naive	Pharmacologic modulation dopaminergic	Formalin score	Chemical (Formalin)	Pharmacologic agent dependent	[96]
				Spinal cFos			
				Spinal electro-physiology			
				Paw withdrawal	Thermal		
Anterior Insula	Rat	Sciatic de-nervation	Pharmacologic modulation dopaminergic	Autotomy score	No stimulus	Decreased score	[97]
Anterior Insula	Rat	Naive		Formalin score	Chemical (formalin)	Decreased score	[97]
Anterior Insula	Rat	Inflam-matory and neuropathic	Lesion	Paw withdrawal	Thermal and Mechanical	Reduced sensitivity	[90]
Anterior and Posterior Insula	Rat	Chronic Pancreatitis	No manipulation	cFos staining	No Stimulus	Increased expression	[91]
			Pharmacologic (CNQX, APV) and chemogenetic inhibition	Abdomen withdrawal	Mechanical	Reduced sensitivity	
Anterior Insula	Mouse	Neuropathic (nerve ligation)	Pharmacologic inhibition (CNQX)	Paw withdrawal	Mechanical	Reduced sensitivity	[92]
Anterior Insula	Rat	Neuropathic (SNI)	fMRI	BOLD signal	No stimulus	Enhanced activation	[94]
Anterior Insula	Mouse	Social transfer of pain	Pyramidal or interneuron cell ablation	Paw withdrawal	Mechanical	Increased and decreased sensitivity respectively	[99]
Anterior Insula	Mouse	Social transfer of pain	CoCl_2_ inactivation	Writhing test	Chemical	Decreased writhing	[100]

## 5. Insular Connectivity to and from Other Pain-Related Brain Regions

The insular cortex integrates sensory information with inputs from other networks. This is also witnessed by the extensive interconnections between the insula and other brain regions. Multiple sensory inputs from different sources get processed within the insula. The pIC receives inputs that emanate from thalamic nuclei [101], some of which are indirect nociceptive and thermoceptive sensory inputs from spinal cord lamina I neurons [25,102]. Indeed, the ventroposterior lateral thalamic and posterior thalamic nuclei, which are involved in pain processing [103], send inputs to the pIC [20,104]. In addition, the pIC recieves processed sensory information from the secondary sosmatosensory cortex [105] as well as from the primary somatosensory cortex [20]. These fit well with the consensually recognized role of the pIC in sensory-discriminative pain coding. The aIC receives thalamic inputs mainly from the medial nuclei of the thalamus [20,101,106]. Integration of these with sensory inputs from ventroposterior thalamic, primary, and secondary somatosensory inputs, in combination with inputs from the amygdala and prefrontal cortex [20], reveals a main role of the aIC in the affective and cognitive functions of pain. Both the pIC and aIC receive inputs from the amygdala [20], mainly from the basolateral amygdala but not from the central nuclei. Other limbic regions that send afferents to the pIC are the anterior cingulate [85] and midcingulate cortex [84]. This is significant, because most human imaging studies find the cingulate and the insular cortex to coactivate in pain [25]; thus, this might imply that pIC activity is downstream of the cingulate activity.

Tracing studies in the mouse have also shown inputs to the aIC from the pIC [20], confirming observations in humans that show aIC activation temporally follows pIC activation [46] and spectral coherence in simultaneous recordings of these subregions [107], which denotes a serial stream of information flow from the posterior to the anterior parts of the insula for pain. This information stream might represent the proposed re-representation of the physiological condition of the body in the aIC from the pIC [25], involved in emotional salient decision making. In addition, both the pIC and the aIC receive cholinergic modulatory inputs from the nucleus basalis of Meynert [34]. These inputs might modulate the interaction between the aIC and pIC and play a role in aversive learning in pain. Alterations of the cholinergic input in the insula in neuropathic pain [88] might dysregulate pIC–aIC information flow and could, consequently, be responsible for the observed shift in insular activation from the posterior to the anterior in chronic pain. Finally, the pIC and aIC receive inputs from the motor cortex [20], with the aIC receiving stronger inputs, which could potentially be involved in motor cortex stimulation-induced analgesia [108].

Many of the inputs to the pIC and aIC are reciprocal connections with the target brain areas, thus forming looped interactions for information processing. These include outputs to the thalamus, the primary and secondary cortex, and the amygdala [20,105,109]. However, the pIC outputs to the amygdala are stronger than those of the aIC, confirming a stronger role in affective pain processing. The insular outputs to the amygdala have also recently been shown to underlie empathic pain in mice [99]. A major output of the aIC is to the striatal nuclei of caudate putamen and nucleus accumbens [20,106,110], indicating a role in the motivational response to pain by engaging the reward/aversion system. In addition, these outputs, in coordination with outputs to the parabrachial [20,111,112] and feedback loops from the motor cortex, might be involved in nocifensive behaviors.

An important function of the insular cortex is mediated through its outputs to the descending pain modulatory pathway. Stimulation of the pIC evokes spinal dorsal horn potentials by a descending pathway that passes through the primary somatosensory cortex [86]. This pathway is proposed to be involved in the maintenance of allodynia during neuropathic pain since pIC lesions alleviate long-term allodynia in rats. An additional descending pathway also connects the pIC to the spinal cord via the raphe magnus nucleus and exerts its pro-algesic action by serotonergic modulation of the spinal cord circuitry [84]. Similarly, activation of the aIC also activates a pro-algesic descending pathway. Indeed, the aIC sends outputs to the raphe magnus nucleus [89], and inhibition of the aIC by increasing its local GABA concentration increases nociceptive heat thresholds in a spinal adrenergic-dependent manner. Thus, since the raphe magnus also connects to the locus coeruleus, the aIC modulates nociception via a raphe magnus–locus coeruleus–spinal cord descending pathway. In neuropathic rats, enhanced functional connectivity between the aIC and the locus coeruleus has also been described using imaging [94]. At the aIC level, this pathway is regulated by opioidergic [95], dopaminergic [96], and oxytocinergic [98] systems. A direct connection of the insular oral region to the spinal trigeminal nucleus of the medulla that facilitates nociceptive and wide-dynamic-range neurons and has pro-nociceptive actions has also been described [113,114]. Additional influence on the descending pain modulatory system can occur by direct pIC outputs to the PAG [115]. The aIC could also indirectly influence descending pathways by outputs to the frontal cortex [20,106,111], which, through top-down descending pathways from the medial prefrontal cortex exerts analgesic control [116].

## 6. Discussion and Conclusions

It is evident that the insular cortex plays a major role in pain, however our knowledge of the way it contributes to pain and its involvement in the chronification of pain is still in its infancy. Clinical and preclinical data show that increased insular activity is linked to increased pain perception and pain hypersensitivity in pathological conditions. Data from animal models suggest that the contribution of the increased insular activity to pain hypersensitivity is through activation of pro-nociceptive descending pathways. This places the insula in a central position on the descending modulatory network (Figure 1). Such a direct connection of the insula with the descending network has not yet been described in humans; nevertheless, studies suggest the involvement of the insula in pain modulation. For example, placebo analgesia was shown to reduce insular activity during a noxious stimulus [117]. Similarly, using a conditioned pain modulation paradigm to study descending pain modulation in healthy subjects, it was shown that weaker pain inhibition is related to increased connectivity between the insula and amygdala [118]. Interestingly, insular transcranial magnetic stimulation in neuropathic patients is analgesic [82]. Similar results have recently been described in rodent models [119,120]. Repeated transcranial current stimulation of the pIC diminishes mechanical pain hypersensitivity in a mouse model of neuropathic pain, an effect which lasts for 2–3 weeks [119]. Similarly, repeated electrical stimulation of the pIC in the rat reduces neuropathic hypersensitivity [120]. At first sight, the antinociceptive effect of the repeated insula stimulation seems contradictory to the prevalent view of increased pain sensation with increased insula activation. This is because the mechanisms by which deep brain stimulation exerts its beneficial effects is poorly understood. Thus, it could involve subareas of the insula or cellular subpopulations (for example, GABAergic neurons); incoming fibers, including neuromodulatory serotonergic, dopaminergic, and cholinergic fibers; or descending outputs. In addition, brain stimulation could affect local plasticity and long-lasting maladaptive changes that take place in chronic pain [121]. Further research will be essential to enrich our understanding on the mechanisms underlying insular contribution to normal and chronic pain, the role of specific cell types, and the alterations in the local circuity that might lead to pain chronification. Future research should also attempt to harmonize insular function with emerging views on pain mechanisms. For example, in such a recent view [122], pain and the nociceptive somatosensory system is proposed to consist of two distinct but interconnected branches, an exteroceptive and an interoceptive branch [123]. The exteroceptive component senses and assesses external threats that might lead to injury and initiates behaviors directed at avoiding them. When this fails and injury occurs, the interoceptive branch takes over, sensing tissue injury and activating protective, caring, and soothing behaviors. It is plausible that the insular cortex could operate at the interface of the two proposed branches since it is receiving and intergrading interoceptive and somatosensory information. It could functionally connect them and allow crosstalk between them, thus mediating the switching of the mental state from one to the other branch. In a slightly different viewpoint [2], two pain states are distinguished: a conscious pain state and non-conscious nociceptive processes. It is suggested that nociception continuously and subconsciously occurs in the absence of pain perception, which serves the avoidance of injury, and that a threshold or set point of nociceptive activity transforms it into conscious pain. Alterations in the threshold during pain chronification lead to pain hypersensitivity and allodynia. As part of the salience network [124], the insula could be involved in the switch from subconscious to conscious pain. Alternatively, the insula could be involved in prediction error [125] calculations between the expected/current threshold and actual sensory information, thus mediating the transformation of the unconscious nociceptive information to conscious pain perception. In this context, alteration of the insular circuitry could update the threshold value to a lower point during chronic pain, leading to pain hypersensitivity.

Similar to most regions involved in pain processing, besides its role in pain, the insula is involved in a multitude of other functions. Therefore, alterations in the insular circuitry stemming from pathologic pain conditions might also affect other insular functions to the extent they share common components. This could constitute the basis for ensuing pain-related comorbidities [126]. Depression, for example, is a common comorbid condition in chronic pain, while in major depressive disorder, the insular cortex shows altered functionality and connectivity [41,127,128]. Conversely, neurological disorders that affect brain areas that also process pain, including the insula, might contribute to pain chronification susceptibility [6]. Investigating the commonality in circuit alterations and mechanisms between pain and comorbid disorders could improve our understanding of chronic pain conditions and increase the available strategies for therapeutic intervention.

## Figures and Tables

**Figure 1 ijms-24-05736-f001:**
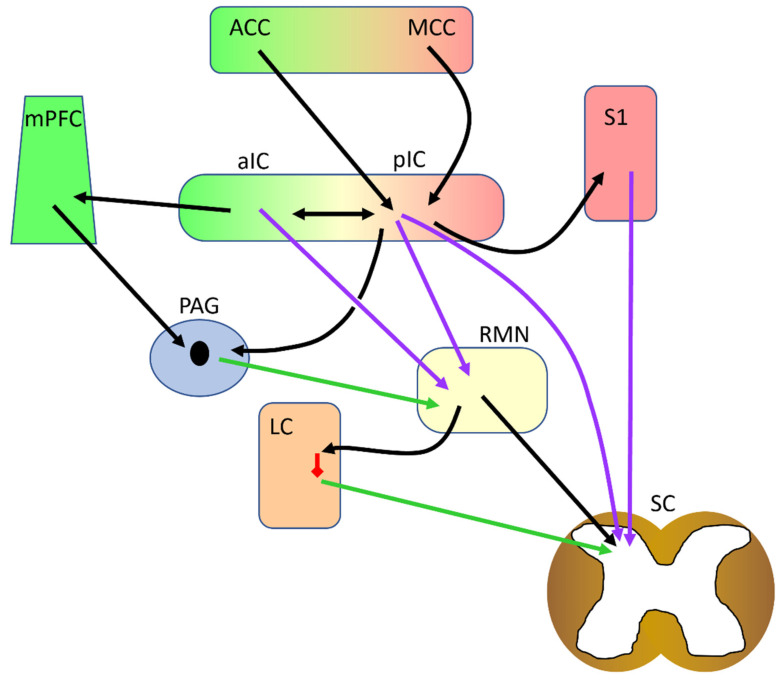
The central role of the insular cortex in the descending modulatory network. Direct and indirect pathways of descending modulation are shown. Green arrows show antinociception, lilac arrows denote pronociceptive action, red shows GABAergic inhibition. ACC: anterior cingulate cortex, aIC: anterior insular cortex, LC: locus coeruleus, MCC: midcingulate cortex, mPFC: medial prefrontal cortex, PAG: periaqueductal gray, pIC: posterior insular cortex, RMN: raphe magnus nucleus, S1: primary somatosensory cortex, SC: spinal cord.

## Data Availability

No new data were created or analyzed in this study. Data sharing is not applicable to this article.

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
