# Peer review of "The Role of the Insular Cortex in Pain"

_ijms, 2023, doi:10.3390/ijms24065736_

Round 1

Reviewer 1 Report

This manuscript is not up to the mark to published. 

Reviewer comments:

There is no info about the structure of insula.

There is no info about how do chemical pathways involve in pain in insula?

There is no info about how do insula correlate pain and other biological function?

There is no info about mechanistic analysis of insula in pain.

There is no info about how do the pain can be mitigated through insula?

There is no future direction from this manuscript.

Overall, I suggest rejecting the manuscript.

Author Response

Response to Reviewer 1 Comments

I thank the Reviewer for comments on the manuscript.

  • There is no info about the structure of insula.

I added a short description of the structure of the insula (page 2, lines 58-62)

  • There is no info about how do chemical pathways involve in pain in insula?

Research on biochemical pathways of the insula that are involved in pain perception are very limited. Most of the research is focused on plasticity mechanism during pain chronicity. I report this and provide a reference to a recent review on the topic (reference 121). Additionally, I describe the possible involvement of Glutamatergic, GABAergic, muscarinic, dopaminergic and opioidergic pathways (page 6).

  • There is no info about how do insula correlate pain and other biological function?

Indeed, there is limited knowledge on the correlation of pain and other biological functions of the insula. I highlight this in the discussion, where I describe the correlation of the involvement of the insula both in pain and depression. I use this as an example of the importance in studying the interaction between pain and other insular functions as a future direction (page 11).

  • There is no info about mechanistic analysis of insula in pain.

At the network level, accumulating evidence and recent research illustrates the descending control network as a mechanism of the insula involvement in pain sensation. I describe and discuss these findings (pages 9-10, Figure 1)

At the cellular level, mechanistic information is mostly available on plastic changes that lead to chronic pain. I include this info in the manuscript (lines 188-194, reference 121).

In addition, I discuss the possible involvement of the insula in light of newer views on pain mechanisms (lines  308-331)

  • There is no info about how do the pain can be mitigated through insula?

Information about treatment options through the insula in pathological pain is very limited. Only recently the option of deep brain stimulation of the insula has been suggested. I describe and discuss this in the manuscript (page 3, lines 137-147 and page 9, lines 293-305)

  • There is no future direction from this manuscript.

In the Discussion section I highlight possible future directions in light of emerging brain mechanisms for pain perception, the interaction of other insular functions and pain comorbidities and deep brain stimulation treatment options

Reviewer 2 Report

The summery is more like an introduction and therefore should be make a modification. That means, it should give a summery about the contents, the main viewpoints, and the conclusive words.

Author Response

Response to Reviewer 2 Comments

I thank the Reviewer for constructive comments on the manuscript.

 - The summery is more like an introduction and therefore should be make a modification.

As per the Reviewers request, I revised the abstract to better reflect the contents of the review

Reviewer 3 Report

This is a very comprehensive literature review focusing on the role of the insular cortex in pain. It is a very interesting paper by a known expert in the field. 

There are some minor concerns, which may detract from the impact and clearness of the manuscript. Specific concerns are listed below:

  • Are there in vitro studies available? Organoids or other 3D models? Please describe.
  • It would be great if the author could add the same table as table 1 but for human studies.
  • Are there any data available on chronic pain conditions which are hereditary and involve the insula? Please describe.
  • For better understanding of the paper it would be great  to make a figure summarizing the different pain types and how the insula is involved respectively.

Author Response

Response to Reviewer 3 Comments

I thank the Reviewer for constructive comments on the manuscript.

  • It would be great if the author could add the same table as table 1 but for human studies.

I have added a table (now Table 1) with an overview of human studies (page 4).

  • Are there in vitro studies available? Organoids or other 3D models? Please describe.

Studies on organoids and 3D models are not available to my knowledge. There are in vitro studies in slice preparations limited mainly to the role of plasticity in pain chronification. These studies are included in the manuscript without however going into great detail. For this topic an extensive review already exists (reference 121) and also is beyond the scope of this review.

  • Are there any data available on chronic pain conditions which are hereditary and involve the insula? Please describe.

To my knowledge, there are no data available on insula related hereditary chronic pain conditions yet.

  • For better understanding of the paper it would be great to make a figure summarizing the different pain types and how the insula is involved respectively.

Current data do not show differential role of the insula for different pain types. I placed the relevant information in the columns “Subject condition” and  “Pain model” within the Tables 1 and 2.

In addition, I have made a correction to figure 1 to include a connection from the insula to the PAG. Changes in the manuscript are with red ink

Round 2

Reviewer 1 Report

The manuscript has been extensively improved as suggested. It can be accepted in the present form.